# Demographic, social, and behavioral correlates of SARS-CoV-2 seropositivity in a representative, population-based study of Minnesota residents

Jordan Abhold[1], Abigail Wozniak[2], John Mulcahy[1], Sara Walsh[3], Evelyn Zepeda[3], Ryan Demmer[1,4], Stephanie Yendell[5], Craig Hedberg[6], Angela Ulrich[6,7], Rebecca Wurtz[8], Timothy Beebe[8]*

1 Division of Epidemiology and Community Health, School of Public Health, University of Minnesota, Minneapolis, MN, United States of America, 2 Opportunity & Inclusive Growth Institute, Federal Reserve Bank of Minneapolis, Minneapolis, MN, United States of America, 3 Health Sciences, NORC at the University of Chicago, Chicago, IL, United States of America, 4 Department of Epidemiology, Mailman School of Public Health, Columbia University, New York, NY, United States of America, 5 Health Risk Intervention Unit, Minnesota Department of Health, St. Paul, MN, United States of America, 6 Division of Environmental Health Sciences, School of Public Health, University of Minnesota, Minneapolis, MN, United States of America, 7 Center for Infectious Disease Research and Policy, Office of the Vice President for Research, University of Minnesota, Minneapolis, MN, United States of America, 7 School of Public Health, University of Minnesota, Minneapolis, MN, United States of America

* beebe026@umn.edu

**Data Availability Statement:** All relevant data are within the paper and its Supporting Information files.

## Abstract

### Background

Monitoring COVID-19 infection risk in the general population is a public health priority. Few studies have measured seropositivity using representative, probability samples. The present study measured seropositivity in a representative population of Minnesota residents prior to vaccines and assess the characteristics, behaviors, and beliefs of the population at the outset of the pandemic and their association with subsequent infection.

### Methods

Participants in the Minnesota COVID-19 Antibody Study (MCAS) were recruited from residents of Minnesota who participated in the COVID-19 Household Impact Survey (CIS), a population-based survey that collected data on physical health, mental health, and economic security information between April 20 and June 8 of 2020. This was followed by collection of antibody test results between December 29, 2020 and February 26, 2021. Demographic, behavioral, and attitudinal exposures were assessed for association with the outcome of interest, SARS-CoV-2 seroprevalence, using univariate and multivariate logistic regression.

### Results

Of the 907 potential participants from the CIS, 585 respondents then consented to participate in the antibody testing (64.4% consent rate). Of these, results from 537 test kits were included in the final analytic sample, and 51 participants (9.5%) were seropositive. The

**Funding:** This study was supported by funding from the Minnesota Department of Health Contract Number 183558.: The study design was completed in advance of funding receipt and was the basis of the awarded contract. Data analysis and manuscript preparation was performed by the author team, which includes a scientist employed by the funder, the Minnesota Department of Health (Dr. Stephanie Yendell), who consulted on epidemiology and state infection trends. The funder (MDH) had no role in the decision to publish.

**Competing interests:** No competing interests.

overall weighted seroprevalence was calculated to be 11.81% (95% CI, 7.30%-16.32%) at of the time of test collection. In adjusted multivariate logistic regression models, significant associations between seroprevalence and the following were observed; being from 23–64 and 65+ age groups were both associated with higher odds of COVID-19 seropositivity compared to the 18–22 age group (17.8 [1.2–260.1] and 24.7 [1.5–404.4] respectively). When compared to a less than $30k annual income reference group, all higher income groups had significantly lower odds of seropositivity. Reporting practicing a number of 10 (median reported value in sample) or more of 19 potential COVID-19 mitigation factors (e.g. handwashing and mask wearing) was associated with lower odds of seropositivity (0.4 [0.1–0.99]) Finally, the presence of at least one household member in the age range of 6 to 17 years old was associated with higher odds of seropositivity (8.3 [1.2–57.0]).

## Conclusions

The adjusted odds ratio of SARS-CoV-2 seroprevalence was significantly positively associated with increasing age and having household member(s) in the 6–17 year age group, while increasing income levels and a mitigation score at or above the median were shown to be significantly protective factors.

## Introduction

In the first year of the COVID 19 pandemic, population-specific SARS-CoV-2 seroprevalence studies were conducted in a variety of groups, such as healthcare and emergency workers [1–6], office workers, children, and pregnant women [7–9], to understand the epidemiology of the disease. These data allowed for more complete case ascertainment than symptomatic diagnostic testing alone and provided important insights into the risk of infection for specific types of exposure.

However, few studies estimated seroprevalence in the general population [6, 10–16]. Several of these general population studies were residual serum studies—testing of serum obtained for other purposes—introducing notable bias [13]. Few measured seropositivity in representative, probability samples of the general population [11, 12, 14, 15]. Fewer still looked beyond sociodemographic characteristics of the sample to characterize the attitudinal and behavioral correlates of infection [6, 17–20]. Finally, most seroprevalence studies were conducted early in the pandemic when the quality of serologic assays was questionable [19].

To address these issues, the present study utilized a representative probability sample of Minnesota residents, collecting data by survey between April 20 and June 8 of 2020 on physical and mental health, economic security, and behavior changes in relation to COVID, followed by serologic testing between late December 2020 and late February 2021. These unique linked data–hereafter referred to as the Minnesota COVID-19 Antibody Study or MCAS–allowed us to measure seropositivity in a representative population prior to vaccines and to assess the characteristics, behaviors, and beliefs of the population and their association with subsequent infection.

## Methods

### Study design and population

Participants in MCAS were recruited by NORC from the residents of Minnesota who participated in the COVID-19 Household Impact Survey (CIS), a statistically valid, population-based

survey that collected data on physical health, mental health, and economic security from the U.S. adult household population nationwide and for 18 regional areas including 10 states (CA, CO, FL, LA, MN, MO, MT, NY, OR, TX) and 8 Metropolitan Statistical Areas (Atlanta, Baltimore, Birmingham, Chicago, Cleveland, Columbus, Phoenix, Pittsburgh) during the early months of the pandemic [20, 21]. Methods for selecting the sample and conducting the surveys are described in Swaziek, *et al*. and Wozniak [20].

In the original CIS, data were collected on basic demographic characteristics, household size and income, current and underlying physical health, mental health, economic security information, and personal behaviors around COVID-19. Data were collected on nineteen mitigation behaviors (including handwashing and mask-wearing but also a wide range of other practices that polls at the time were tracking such as whether or not participants had prayed, stockpiled food and water, wiped packages entering their homes, avoided high risk people, avoided some or all restaurants, and cancelled or postponed various activities).

For the purposes of the MCAS follow-up study, age in the CIS data was collapsed into 3 categories (18–22, 23–64, older than 64), race/ethnicity variables were collapsed into two categories (non-white and white), and education level was categorized as less than college degree, associates/bachelor's degree, and post-grad degree. Self-reported general health status was categorized as excellent/very good or good/fair/poor. COVID-related symptoms at the time of the initial survey and a history of any of the CDC-listed risk factors for severe COVID-19 at the time of the CIS (diabetes, high blood pressure or hypertension, heart disease, heart attack or stroke, asthma, chronic lung disease and COPD, bronchitis and emphysema, allergies, a mental health condition, cystic fibrosis, liver disease or end stage liver disease, cancer, a compromised immune system, and overweight or obesity) were each categorized as "yes" or "no." Questions in the CIS survey which screened for depression were combined to indicate the presence or absence ("yes" or "no") of any poor mental health days. The CIS survey asked about 19 behaviors that people may engage in to reduce their risk of acquiring COVID-19. We incorporated these questions into our analysis in two ways. First, we dichotomized the total count of reported behaviors around the sample median of 10 reported behaviors. We also examined if respondents engaged in specific behaviors of masking and social distancing, which have since been shown to meaningfully reduce COVID risk [22, 23]. We have termed this variable the "Big 2", and it is coded as reporting masking and social distancing, reporting one of the two, or reporting neither. Finally, a variable was constructed to indicate the presence of children of various ages in the household compared to only adults.

## Serosurvey

The Minnesota CIS sample surveyed 1,071 unique respondents, of whom 912 consented to be re-contacted and 907 provided complete contact information, including email addresses. This group served as the potential participants in the serosurvey, and NORC sent recruitment emails to these individuals with a web link and unique PIN offering participation in the antibody testing program. Respondents were pointed to a consent portal where they would sign up to receive a capillary blood collection kit mailed to their home, to self-collect capillary blood using Neoteryx Mitra® 10 μl samplers by volumetric absorption of were offered a $25 incentive as well as their antibody test results for participation. Of the 907 potential participants, 585 respondents then consented to participate in the antibody testing (64.4% consent rate). Of these, 581 were sent kits, and 540 test kits were returned, and results from 537 test kits with complete survey data were included in the final analytic sample. Specimens were then tested using the Quansys Q-Plex™ SARS-CoV-2 Human IgG (Quansys Biosciences, Logan, UT) [24].

## Study variables

The primary outcome of interest in this study was SARS-CoV-2 seropositivity. The primary exposures of interest were age, sex, race/ethnicity (white/non-white), income, population density of place of residence (defined as rural, suburban, or urban), education level, household make-up, at/above median mitigation score, "Big 2" score, and poor mental health days (yes/no).

## Statistical analysis

Prior to analysis of the CIS data, an iterative raking process was used to adjust for any survey nonresponse as well as any non-coverage or under and oversampling. Raking variables included age, gender, race/ethnicity, education, and county groupings based on county level counts of the number of COVID-19 deaths. Demographic weighting variables were obtained from the 2018 American Community Survey. The weighted data reflect the population of adults aged 18 and over in each region. The overall weighted seroprevalence was adjusted for testing error, using the following formula [25] below, sensitivity/specificity estimates from Quansys3027, and the methodology described by Demmer, et al. [26]:

$$adjusted\ prevalence = \frac{crude\_prevalence + specificity - 1}{sensitivity + specificity - 1}$$

SAS version 9.4 was used for statistical analyses, including univariate descriptive statistics, univariate logistic regressions assessing the association between seropositivity and each factor of interest, as well as multivariate logistic models for each variable of interest adjusted for the age, sex, income, population density, simplified education level, household make-up, at/above median mitigation score, and poor mental health days (yes/no) reported by participants.

Informed consent was obtained electronically within the survey for participants in the CIS, electronically via an online web portal for MCAS participants, and the study was approved by the University of Minnesota Institutional Review Board (#STUDY00011527).

## Results

Table 1 summarizes the characteristics of the study sample as well as unadjusted seroprevalence of the sample according to key variables of interest. The unadjusted seropositivity rate for the study population was 9.5% (51 out of the 537 returned test kits). The weighted and adjusted rate was 11.81% (95% CI, 7.30%-16.32%). Weighted seroprevalence varied by population density, ranging from 20.81% (95% CI, 4.23%-37.39%) in rural areas to 11.92% (95% CI, 6.24%-17.59%) in urban areas to 3.78% (95% CI, 0.57% -6.99%) in suburban areas. Weighted seroprevalence rates were observed to be at least five percentage points higher than the population rate for males (13.77%; 95% CI, 6.61% - 20.92%), individuals with 1–2 children in the household (19.78%; 95% CI, 5.72% - 33.83%), and those with school-age children between 6 and 17 years (21.41%; CI, 7.65% -35.18%).

In the multivariate analyses (Table 2), the following demographic variables were significantly related to a higher likelihood of seropositivity: 1) age 23–64 years (OR = 17.79; 95% CI, 1.22–260.08) compared to the 18–22 years group; 2) age 65 years or more (OR = 24.68; 95% CI, 1.51–404.44) compared to the 18–22 years group; and 3) and respondents reporting having school-age children (aged 6–17 years) in the household (OR = 8.253; 95% CI, 1.20–56.98) compared to not having children in this age group. Factors that decreased the likelihood of seropositivity include earning between $30,000 - $60,000 per year (OR = 0.20; 95% CI, 0.05–0.91) and earning more than $125,000 per year (OR = 0.14; 95% CI, 0.02–0.76) compared to those earning less than $30k. None of the potential risk factors–good/fair/poor health status, COVID-related symptoms at the time of the survey, presence of high-risk health conditions,

**Table 1. SARS-CoV-2 seroprevalence by selected variables and seropositivity associations with various demographic, risk, and protective factors.**

| Variable | Total participants | Seropositive participants No. (%) | Weighted % (95% CI) |
|---|---|---|---|
| Minnesota Overall | 537 | 51 (9.5%) | 11.81% (7.30–16.32%)[a] |
| *Demographics* | | | |
| Population density | | | |
| Urban | 386 | 34 (8.81%) | 11.92% (6.24–17.59%) |
| Suburban | 97 | 9 (9.28%) | 3.78% (0.57–6.99%) |
| Rural | 54 | 8 (14.81%) | 20.81% (4.23–37.39%) |
| Age | | | |
| 18–22 | 13 | 1 (7.69%) | 1.80% (0.00–5.48%) |
| 23–64 | 369 | 38 (10.33%) | 11.75% (6.71–16.79%) |
| 65+ | 155 | 12 (7.74%) | 12.58% (0.94–24.23%) |
| Sex | 536 (1 missing) | 51 | |
| Male | 221 | 21 (9.50%) | 13.77% (6.61–20.92%) |
| Female | 315 | 30 (9.52%) | 9.34% (3.77–14.92%) |
| Simplified Race | | | |
| White | 372 | 39 (10.48%) | 11.63% (6.31–16.95%) |
| Non-white | 165 | 12 (7.27%) | 11.20% (3.17–19.23%) |
| Household income | 526 (11 missing) | 51 | |
| Less than $30K | 41 | 5 (12.20%) | 23.60% (2.86–44.34%) |
| $30K to < $60K | 116 | 12 (10.35%) | 9.91% (1.86–17.96%) |
| $60K to < $125K | 223 | 20 (8.97%) | 8.76% (3.20–14.32%) |
| $125K + | 146 | 14 (9.59%) | 11.04% (4.27–17.81%) |
| Simplified Education | | | |
| Less than college degree | 107 | 12 (11.22%) | 11.07% (3.62–18.51%) |
| Associates/bachelor's degree | 263 | 30 (11.41%) | 13.64% (7.33–19.95%) |
| Post-grad degree | 167 | 9 (5.39%) | 7.95% (1.20–14.71%) |
| Household size | | | |
| Alone | 156 | 12 (7.69%) | 9.45% (1.42–17.48%) |
| 1+ other adult(s) | 266 | 24 (9.02%) | 9.36% (3.40–15.32%) |
| 1 or 2 children | 87 | 9 (10.35%) | 19.78% (5.72–33.83%) |
| ≥ 3 children | 28 | 6 (21.43%) | 12.01% (0.17–23.84%) |
| Binary of household member(s) w/in age range | n/a | | |
| Aged 0–5 years | 52 | 4 (7.69%) | 9.18% (0.00–20.03%) |
| Aged 6–17 years | 84 | 13 (15.48%) | 21.41% (7.65–35.18%) |
| Aged 18 + years (*not counting participant) | 372 | 38 (10.22%) | 12.37% (6.82–17.92%) |
| *Risk Factors* | | | |
| General health status | | | |
| Excellent/very good | 376 | 36 (9.57%) | 9.48% (4.88–14.09%) |
| Good/fair/poor | 161 | 15 (9.32%) | 14.76% (5.58–23.95%) |
| COVID-related symptoms [b] | | | |
| Yes | 365 | 29 (7.95%) | 10.97% (5.08–16.86%) |
| No | 172 | 22 (12.79%) | 12.50% (5.87–19.13%) |
| High-risk health condition [c] | | | |
| Yes | 428 | 40 (9.35%) | 11.92% (6.52–17.32%) |
| No | 109 | 11 (10.09%) | 9.81% (2.51–17.11%) |
| Poor mental health days reported | | | |
| Yes | 392 | 39 (9.95%) | 13.70% (7.69–19.71%) |
| No | 141 | 12 (8.51%) | 6.02% (1.62–10.43%) |

*(Continued)*

**Table 1.** (Continued)

| Variable | Total participants | Seropositive participants No. (%) | Weighted % (95% CI) |
|---|---|---|---|
| *Protective Factors* | | | |
| Mitigation Score (median = 10) | | | |
| Below median | 252 | 34 (13.49%) | 14.54% (7.85–21.23%) |
| At or above median | 285 | 17 (5.96%) | 7.54% (1.81–13.28%) |
| Big 2[d] | | | |
| Both | 436 | 36 (8.26%) | 11.66% (6.07–17.24%) |
| Neither | 101 | 15 (14.85%) | 10.88% (4.09–17.67%) |

[a] Adjusted using methodology referenced above.

[b] Includes fever, chills, runny or stuff nose, chest congestion, skin rash, cough, sore throat, sneezing, muscle or body aches, headaches, fatigue or tiredness, shortness of breath, abdominal discomfort, nausea or vomiting, diarrhea, changed or lost sense of taste or smell, and/or loss of appetite in past 7 days.

[c] Includes diabetes, high blood pressure or hypertension, heart disease, heart attack or stroke, asthma, chronic lung disease or COPD, bronchitis or emphysema, allergies, a mental health condition, cystic fibrosis, liver disease or end stage liver disease, cancer, a compromised immune system, overweight or obesity.

[d] "Big 2" referring to mask wearing and social distancing

or poor mental health–departed meaningfully from the overall statewide seroprevalence rate (Table 1) or attained statistical significance in the multivariate analyses (Table 2).

Our examination of possible protective factors, namely engaging in personal public health-oriented behaviors such as postponing work-related activities or avoiding public or crowded places, showed that those who engaged in less than the median number of those efforts had higher rates of seropositivity than what was observed at the statewide level (14.54%; 95% CI, 7.85 - 21.23%: see Table 1). In the multivariate model (Table 2), engaging in higher than the median number of protective/mitigation behaviors lowered the chances of becoming seropositive (OR = 0.36; CI, 0.23–0.99).

## Discussion

The MCAS study links demographic and behavioral data from early in the COVID-19 pandemic to serostatus six months later. During this 6-month period, the State of Minnesota reached a peak reported seropositivity rate of 17.2% in November 2020 [27]. The overall MCAS seropositivity rate (11.81%) was lower than the estimated seroprevalence of 15.9% (95% CI, 13.3%-18.6%) observed for Minnesota in the CDC nationwide commercial laboratory survey in the first half January 2021 [27]. During this 6-month period, Minnesota also experienced rises in hospitalizations and deaths that paralleled the experiences of many other parts of the US.

Three observations were robust. First, our multi-variate regression found that older age groups were associated with higher odds of seropositivity and that living in a higher income household was associated lower odds of seropositivity. A recent COVID-19 seroprevalence study conducted in the city of Belém, Brazil also found older age and lower income to be associated with seropositivity in the early waves of the pandemic [28]. These parallel findings suggest age and income impact risk of seropositivity even in distinct cultural and economic settings, and further demonstrate socio-economic disparities in COVID-19 risk [29, 30].

Our data also show that adults with school-aged children in their household had more than eight times the odds of seroprevalence after adjusting other variables. The significance and magnitude of the association between seropositivity and living with children of school age suggest that there are COVID-19 risk factors associated with the circumstances that accompany raising children. This may support the fact that school-age children can become infected and transmit SARS-CoV-2 infections and contribute to family and community spread; however,

**Table 2. SARS-CoV-2 crude and adjusted[a] odds ratios of seropositivity by selected variables[1].**

| Variable | OR (95% CI) Univariate model | OR (95% CI) Multivariate model |
| --- | --- | --- |
| *Demographics* | | |
| Population density | | |
| Urban | Reference | Reference |
| Suburban | 0.29 (0.10–0.82) | 0.381 (0.125–1.161) |
| Rural | 1.94 (0.62–6.10) | 1.675 (0.555–5.055) |
| Age | | |
| 18–22 | Reference | Reference |
| 23–64 | 7.24 (0.86–61.29) | **17.794 (1.217–260.083)** |
| 65+ | 7.83 (0.76–80.85) | **24.682 (1.506–404.440)** |
| Sex | | |
| Male | 1.55 (0.63–3.79) | 2.067 (0.902–4.735) |
| Female | Reference | Reference |
| Race | | |
| White | 1.04 (0.40–2.73) | 1.207 (0.454–3.206) |
| Non-white | Reference | Reference |
| Household income | | |
| Less than $30K | Reference | Reference |
| $30K to < $60K | 0.36 (0.08–1.54) | **0.203 (0.045–0.914)** |
| $60K to < $125K | 0.31 (0.08–1.20) | **0.141 (0.033–0.604)** |
| $125K + | 0.40 (0.11–1.54) | **0.137 (0.024–0.763)** |
| Education | | |
| Less than college degree | Reference | Reference |
| Associates/bachelor's degree | 1.27 (0.50–3.21) | 1.615 (0.686–3.803) |
| Post-grad degree | 0.69 (0.21–2.30) | 0.791 (0.213–2.944) |
| Household size | | |
| Alone | Reference | Reference |
| 1+ other adult(s) | 0.99 (0.31–3.21) | 1.090 (0.381–3.117) |
| 1 or 2 children | 2.36 (0.65–8.62) | 4.603 (0.932–22.734) |
| ≥ 3 children | 1.31 (0.30–5.66) | 2.409 (0.524–11.076) |
| Binary of household member(s) in age range | | (Ref. = absence of each binary) |
| Aged 0–5 years | 0.76 (0.20–2.91) | 0.321 (0.051–2.018) |
| Aged 6–17 years | **2.74 (1.04–7.25)** | **8.253 (1.195–56.984)** |
| Aged 18 + years (*not counting participant) | 1.39 (0.49–3.95) | 3.391 (0.205–56.041) |
| *Risk Factors* | | |
| General health status | | |
| Excellent/very good | Reference | Reference |
| Good/fair/poor | 1.65 (0.67–4.09) | 0.712 (0.296–1.717) |
| COVID-related Sx [bv] | | |
| Yes | 0.86 (0.37–2.03) | 0.766 (0.326–1.798) |
| No | Reference | Reference |
| High-risk health conditions [c] | | |
| Yes | 1.24 (0.47–3.29) | 0.660 (0.254–1.715) |
| No | Reference | Reference |
| Poor mental health days Reported | | |
| No poor mental health days reported | Reference | Reference |
| Any poor mental health days reported | 2.48 (0.98–6.28) | 2.202 (0.900–5.392) |

*(Continued)*

**Table 2.** (Continued)

| Variable | OR (95% CI) Univariate model | OR (95% CI) Multivariate model |
|---|---|---|
| *Protective Behaviors* | | |
| Median mitigation score (median = 10) | | |
| Below median | Reference | Reference |
| At or above median | 0.48 (0.18–1.28) | **0.357 (0.128–0.994)** |
| "Big 2"[d] | | |
| Both | Reference | Reference |
| Neither | 0.93 (0.38–2.25) | 1.121 (0.472–2.665) |

[1]Bolding in Table 2 indicates a statistically significant finding with a p-value > 0.05 for the parameter estimate of odds ratio/adjusted odds ratio of seroprevalence.

[a] Multivariate logistic models were adjusted for age, sex, income, population density, simplified education level, household make-up, at/above median mitigation score, and poor mental health days (yes/no)

[b] Includes fever, chills, runny or stuff nose, chest congestion, skin rash, cough, sore throat, sneezing, muscle or body aches, headaches, fatigue or tiredness, shortness of breath, abdominal discomfort, nausea or vomiting, diarrhea, changed or lost sense of taste or smell, and/or loss of appetite in past 7 days.

[c] Includes diabetes, high blood pressure or hypertension, heart disease, heart attack or stroke, asthma, chronic lung disease or COPD, bronchitis or emphysema, allergies, a mental health condition, cystic fibrosis, liver disease or end stage liver disease, cancer, a compromised immune system, overweight or obesity.

[d] "Big 2" referring to mask wearing and social distancing

our observational findings cannot determine whether this is the specific driving force of our findings, or whether some other feature associated with having school-aged children made respondents to the survey more vulnerable to the spread of COVID-19. Notably, the largest school districts in Minnesota were operating fully remotely for the period of our study. Further research should be conducted to determine if school-aged children are meaningful contributors to transmission, to inform whether broader testing efforts in schools might be a useful tool to identify infectious individuals, prevent outbreaks, and protect vulnerable members of the community.

The second robust finding relates to behavior intended to mitigate personal and collective risk. A mitigation score at or above the median (engaging in more than 10 of these behaviors) was associated with an adjusted odds ratio of 0.357 (95% CI, 0.128–0.994), indicating that those who–early on–took the pandemic more seriously or changed more of their behaviors were less likely to test positive for the presence of antibodies. While not all of these mitigation measures directly affected one's likelihood of infection, this metric seems to have captured a more general attitude.

Although this study has several strengths, including its deployment of state-of-the-art survey methods, its probability-based sampling approach, the temporal nature of the study design, the unprecedented inclusion of a wide array of social, behavioral, and attitudinal correlates of infection, and the use of a de-centralized capillary blood data collection protocol with high fidelity, it is important to note some potentially important limitations.

First, some racial and ethnic groups were underrepresented in the study relative to the nation as a whole, reflecting the population demographics of Minnesota. Second, the participants represent a group of individuals inclined to participate in studies such as this given their participation in the COVID-19 Household Impact Survey and consent to be re-contacted. These may be people generally inclined to engage in various other forms of prosocial behavior such as mask wearing and social distancing, suggesting the possibility of confounding by indication. Third, COVID-19 vaccines were being made available to healthcare workers and other

high priority groups during the blood sample collection phase of the study. Respondents did not report whether or not they had been vaccinated when returning their samples, so it is not possible to control for or adjust our samples for potential vaccination. However, the impact of early vaccine access on our results is likely to be small, since access was highly restricted during this period and many individuals with vaccine access had only received one dose, which has been shown to be unlikely to generate a seropositive result [31]. The one exception is the population 65 and older, where access had risen such that a quarter of the population had completed the 2-dose vaccine series by the end of our collection period, up from 0.3% at the collection midpoint. We therefore interpret the positive association between older age and seropositivity in our results with some caution, as this is the one dimension in which vaccine access may have generated positivity, in addition to infection. Fourth, respondents did not report whether they had been formally diagnosed with COVID-19 between the baseline survey and bio-sample collection, so we cannot distinguish between previously detected and unreported cases. Fifth, the sample sizes involved often yielded wide confidence intervals at the more granular cuts at the data. Sixth, despite extensive questions on demographic and behavioral COVID-19 risk factors, some factors that could meaningfully impact seroprevalence including substance use before and during the pandemic and occupation in high-risk fields (e.g. healthcare and service workers) were not asked about in CIS. Seventh, we assumed all positive and negative COVID-19 cases as being true cases. In reality, the Human IgG (4-Plex) assay has a reported sensitivity of 97% and specificity of 100% [23]. This imperfect sensitivity along with our long study window may be biasing our seroprevalence estimates towards a lower value. Finally, we acknowledge that many of the variables, such as symptoms, mental health, and personal and public health mitigation behaviors likely varied between the time of the initial survey and the subsequent blood specimen collection. We are in the process of fielding a follow-up study that deploys a design that supports a more contemporaneous assessment.

## Conclusion

Pairing data on pandemic attitudes and behaviors with serologic results provides the most complete insight into transmission risk factors to inform further epidemiologic study. Important risk factors such as school-age children in the household and protective factors such as personal mitigation behaviors suggest that public health planners should focus on these issues as they deal with the current outbreak and those that may emerge in the future.

## Supporting information

**S1 File.**
(CSV)

**S2 File.**
(XLSX)

## Acknowledgments

*Thank you*: We are also profoundly grateful for the study participants who have donated valuable time to advance our understanding about SARS-CoV-2 seroprevalence in Minnesota.

## Author Contributions

**Conceptualization:** Abigail Wozniak, Sara Walsh, Ryan Demmer, Stephanie Yendell, Rebecca Wurtz, Timothy Beebe.

**Data curation:** Sara Walsh, Ryan Demmer, Stephanie Yendell, Timothy Beebe.

**Formal analysis:** Abigail Wozniak, John Mulcahy, Sara Walsh, Ryan Demmer, Stephanie Yendell, Angela Ulrich, Timothy Beebe.

**Funding acquisition:** Abigail Wozniak, Ryan Demmer, Stephanie Yendell, Timothy Beebe.

**Investigation:** Stephanie Yendell, Rebecca Wurtz, Timothy Beebe.

**Methodology:** Jordan Abhold, Abigail Wozniak, John Mulcahy, Sara Walsh, Ryan Demmer, Stephanie Yendell, Rebecca Wurtz, Timothy Beebe.

**Project administration:** Sara Walsh, Evelyn Zepeda, Stephanie Yendell, Timothy Beebe.

**Resources:** Abigail Wozniak, Timothy Beebe.

**Supervision:** Stephanie Yendell, Timothy Beebe.

**Validation:** Jordan Abhold, Timothy Beebe.

**Visualization:** Jordan Abhold.

**Writing – original draft:** Jordan Abhold, Abigail Wozniak, John Mulcahy, Sara Walsh, Evelyn Zepeda, Ryan Demmer, Stephanie Yendell, Craig Hedberg, Angela Ulrich, Rebecca Wurtz, Timothy Beebe.

**Writing – review & editing:** Jordan Abhold, Abigail Wozniak, John Mulcahy, Sara Walsh, Evelyn Zepeda, Ryan Demmer, Stephanie Yendell, Craig Hedberg, Angela Ulrich, Rebecca Wurtz, Timothy Beebe.

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
