## [Decision Letter · Decision Letter 0]

25 Apr 2022

PONE-D-22-04448Demographic, social, and behavioral correlates of SARS-CoV-2 seropositivity in a representative, population-based study of Minnesota residentsPLOS ONE

Dear Dr. Beebe,

Thank you for submitting your manuscript to PLOS ONE. After careful consideration, we feel that it has merit but does not fully meet PLOS ONE’s publication criteria as it currently stands. Therefore, we invite you to submit a revised version of the manuscript that addresses the points raised during the review process.

Both reviewers agree that the study is interesting and timely, but raised important concerns that should be addressed. There are many such concerns, but the most important of these would be when during the year-long study vaccination was rolled out, and whether there might be correlations between those found with antibodies and when they might have been vaccinated. Naturally, please address all concerns raised by the reviewers.

We look forward to receiving your revised manuscript.

Kind regards,

Siew Ann Cheong, Ph.D.

Academic Editor

PLOS ONE

Journal Requirements:

a) Did participants provide their written or verbal informed consent to participate in this study?

“This study was supported by funding from the Minnesota Department of Health Contract Number 183558.”

“This study was supported by funding from the Minnesota Department of Health Contract Number 183558.”

“This study was supported by funding from the Minnesota Department of Health Contract Number 183558.”

6. Please include a copy of Table 13 which you refer to in your text on page 14.

Reviewers' comments:

Reviewer's Responses to Questions

**Comments to the Author**

1. Is the manuscript technically sound, and do the data support the conclusions?

Reviewer #1: Yes

Reviewer #2: Partly

2. Has the statistical analysis been performed appropriately and rigorously? 

Reviewer #1: Yes

Reviewer #2: I Don't Know

3. Have the authors made all data underlying the findings in their manuscript fully available?

Reviewer #1: Yes

Reviewer #2: Yes

4. Is the manuscript presented in an intelligible fashion and written in standard English?

Reviewer #1: Yes

Reviewer #2: Yes

5. Review Comments to the Author

Reviewer #1: This paper examines the socio-demographic and behavioral correlates of SARS-CoV-2 seropositivity in a representative sample of Minnesota residents.

The strengths of the study are its longitudinal design, use of an objective biological marker to identify infection (as opposed to studies of self-reported infection or hospitalization), and the consideration of factors beyond simple differences in age and gender.

There are certain points on which I believe the manuscript would benefit from clarification or correction by the authors, which are listed below:

1. Was mass immunization initiated in Minnesota during the period prior to or during the collection of the seroprevalence samples? (If not, this could simply be stated in the paper. If so, how would this affect the results of antibody tests? Were any questions asked regarding immunization status prior to sampling?)

2. What was the rationale for the division of age categories into 18-22, 23-64 and >65? Do these age groups differ significantly in terms of social mobility (for example is >65 the conventional "retirement age"?) and hence in terms of risk of exposure to COVID-19? Would further subdivisions within the second of these categories (say 23-44 and 45-64) have allowed a more precise analysis of the data?

3. Was any information collected on substance use (alcohol, tobacco, etc.) by study participants? (If not, this could be mentioned among the limitations of the study, as this may constitute a risk factor for infection, and also because there are several reports of increased substance use during periods of isolation or "lockdown").

4. Were any of the study participants employed in a field that entails greater exposure to COVID-19? (This would include not just healthcare workers but other "frontline" workers, such as law enforcement officers, or those involved in the service industries). This would constitute an independent risk factor for SARS-CoV-2 infection; if information on this is available, it could be included in the analyses.

5. What was the epidemiological status of the concerned state / region at the time the study was conducted? Was sampling done during a local "wave" or "spike" of SARS-CoV-2 cases, or during a period of relative quiescence? This could affect the interpretation of the study results. Even if a formal statistical analysis is not carried out, some basic epidemiological details could be provided based on data from local health authorities.

6. Did any study participants develop overt COVID-19 (diagnosed via antigen testing or RT-PCR) in the period between baseline data collection and subsequent antibody testing? This might be of interest when analyzing the frequency of symptomatic vs asymptomatic infection and their respective correlates (for example, was asymptomatic infection more common in younger subjects, or symptomatic infection more common in the elderly?)

7. The Discussion section could be improved by a mention of the age-, income- and location-related differences identified in this study. Though they may be seen as a replication of earlier results, these findings remain of value in terms of larger-scale social and public health interventions to minimize the risk of infection (e.g. by providing financial assistance or minimizing overcrowding in housing).

8. The finding regarding school-age children is interesting, but it cannot be taken as definitive without a knowledge of the children's antibody / antigen status. There are additional explanations that need to be taken into account (e.g. an association between children in the house and lower income; greater potential exposure to infection by parents / caregivers due to the need to purchase essential goods for their children). The wording regarding this finding could be "softened" a little and the need for replication (and for the exclusion of alternate explanations) could be discussed towards the end of the paper.

9. What was the reported sensitivity / specificity of the antibody tests themselves? Was this data available, and how would it affect the confidence which one could place in the final study results?

Overall, I found that this paper had considerable merit and could be further improved by addressing the relatively minor issues mentioned above; I commend the authors for their work.

Reviewer #2: This paper surveys and tests a representative sample of Minnesota residents to identify correlates of SARS-CoV-2 infection risk. The authors take into account a wide range of demographic, social and behavioral factors that could potentially play a role in this context. They conduct the survey in 2020, before vaccines were available, and do the serological tests in early 2021 and conclude that having school-aged children and engaging in numerous mitigation behaviors are related to an increased infection risk.

The research question and participant sample are very interesting and topical. Yet, some methodological choices and how the paper itself is written could potentially be improved.

***General comments:

-The time period between elicited behavioral and social factors and the infection test is quite long, almost one year. Could the authors cite some sources that show how these factors have been relatively stable over time, even though the research findings and the respective policies had changed quite substantially over this same time period?

-Although the range of the elicited behavioral covariates is substantive, it appears that one important variable is not controlled for, namely whether the participants had already been tested positive before and/or whether they had been vaccinated. (Or perhaps I missed it and these people were excluded from the study?) Having been tested positive or getting vaccinated could have behavioral effects, e.g. engaging in fewer mitigating activities due to the newly gained (temporary) immunity.

-The authors mention that similar datasets were gathered for U.S. adult household population nationwide and for 18 regional areas including 10 states and 8 Metropolitan Statistical Areas. If possible, could the authors mention to what extent their Minnesota-specific results replicate elsewhere in the U.S.?

***Specific comments:

*Abstract & Introduction:

-This following paragraph in the abstract is a bit difficult to read and interpret. Please reconsider whether it does justice to your results in the summary, or whether perhaps it is more suited for the detailed results section and a more intuitively worded results summary could be drafted instead.

“In adjusted multivariate logistic regression models, significant associations between seroprevalence and the following were observed; participants’ age categories (17.794 [1.217-260.083] for 23-64 years and 24.682 [1.506-404.440] for 65+ years compared to the 18-22 years category); income category (0.203 [0.045-0.914] for $30k to <$60k, 0.141 [0.033- 0.604] for $60k to <$125k, and 0.137 [0.024-0.763] for $125k+ compared to the <$30k category); and a binary variable for the presence of household member(s) in the age range of 6 to 17 years old (8.253 [1.195-56.984] compared to those who did not have household member(s) in this age group).»

-Please consider introducing a brief definition of the “mitigation score” in the abstract already.

*Methods:

-It is somewhat unclear how the various variables of interest were chosen to be included in the study, i.e. what is the theoretical basis and/or previous evidence for each of them.

-Is it possible to say to what extent do the results hold if slightly different approaches to collapsing the variables are chosen? It appears that some definitions of the variables differ from the ones described in Swaziek et al. (2020) and Wozniak (2020). For example, it is not intuitively clear why the 3 very different age categories (18-22 with only 13(!) participants, 23-64 with 369 participants, older than 64 with 155 participants) are used. Interpretation of these variables can thus be somewhat problematic, as also potentially demonstrated by the differing significance levels between the two logit models presented in Table 2.

-Would it be possible to include more information about the extent to which the variables described in the “Study design and population” and “Study variables” subsections differ and the reasons behind this difference? For example, it is not clear how the mitigation behaviors (as measured by the Big 2 and median of 10 (other?) mitigation behaviors) are related to the data collected on “nineteen mitigation behaviors” mentioned earlier. In addition, it would be interesting to see which of the mitigation behaviors “work better” for preventing infections than others, rather than just examining the absolute number of these behaviors. In fact, the paper does mention this in passing in the Discussion: “While not all of these mitigation measures directly affected one’s likelihood of infection, this metric seems to have captured a more general attitude.»

-Finally, it is also unclear why the wide ranging answers of good/fair/poor health status are combined together in a binary variable. Have the authors considered consistently using mean/median splits for binary classifications like these?

*Results:

-It could be interesting to see whether the results hold in multivariate OLS regressions with standardized variables as well. Although logit regressions are indeed the most suitable approach for the given data, OLS regressions are often more intuitive to interpret. It is therefore (unofficially) a common practice to report both logit and OLS results in these cases, simply to facilitate the ease of reading. That being said, I leave this fully to the authors to decide whether this is something they would like to consider.

*Discussion & Conclusions:

-The authors state that only two results are robust (having school-aged children and engaging in numerous mitigation behaviors), yet the paper stars with also mentioning age and income effects in the abstract. Please adjust the messaging to be consistent throughout the paper accordingly. (In fact, some other studies have shown the age effects to be in the other direction than the less-robust ones reported in this study, with younger people more likely to be infected and older people being more careful in this respect.)

-The sections seem to lack a discussion that puts the paper’s results into the broader context of the vast number of other related studies. How do the paper’s results add to the literature? How do the results relate to other studies?

-Finally, the readers of the paper would greatly benefit from more elaborate instructions for public health planners. What are the policy implications of the results? What should policy makers do (differently) after reading the paper?

6. PLOS authors have the option to publish the peer review history of their article (what does this mean?). If published, this will include your full peer review and any attached files.

Reviewer #1: **Yes: **Ravi Philip Rajkumar

Reviewer #2: No

---

## [Author Response · Author response to Decision Letter 0]

8 Jul 2022

A detailed response to reviewer comments is provided in the the "Response to Reviewers PONE-D-22-04448 FINAL" document that was uploaded into this system.

---

## [Decision Letter · Decision Letter 1]

13 Dec 2022

Demographic, social, and behavioral correlates of SARS-CoV-2 seropositivity in a representative, population-based study of Minnesota residents

PONE-D-22-04448R1

Dear Dr. Beebe,

We’re pleased to inform you that your manuscript has been judged scientifically suitable for publication and will be formally accepted for publication once it meets all outstanding technical requirements.

Kind regards,

Siew Ann Cheong, Ph.D.

Academic Editor

PLOS ONE

Additional Editor Comments (optional):

Reviewers' comments:

Reviewer's Responses to Questions

**Comments to the Author**

1. If the authors have adequately addressed your comments raised in a previous round of review and you feel that this manuscript is now acceptable for publication, you may indicate that here to bypass the “Comments to the Author” section, enter your conflict of interest statement in the “Confidential to Editor” section, and submit your "Accept" recommendation.

Reviewer #1: All comments have been addressed

Reviewer #3: All comments have been addressed

2. Is the manuscript technically sound, and do the data support the conclusions?

Reviewer #1: Yes

Reviewer #3: Yes

3. Has the statistical analysis been performed appropriately and rigorously? 

Reviewer #1: Yes

Reviewer #3: Yes

4. Have the authors made all data underlying the findings in their manuscript fully available?

Reviewer #1: Yes

Reviewer #3: Yes

5. Is the manuscript presented in an intelligible fashion and written in standard English?

Reviewer #1: Yes

Reviewer #3: Yes

6. Review Comments to the Author

Reviewer #1: The revisions made by the authors are satisfactory and thorough in my opinion. I have no further major changes or corrections to suggest.

Reviewer #3: I would like to suggest the authors compare their results with a recent report published in a population living in the Brazilian Amazon region, that showed similar results in that community. I do believe that the comparison with a population located in the South America can be interesting to the readers realize the impact of COVID-19 in population with distint socio-economic reality.

Silva Torres et al. Changes in the seroprevalence and risk factors between the first and second waves of COVID-19 in a metropolis in the Brazilian Amazon. Front. Cell. Infect. Microbiol. 12:932563. doi: 10.3389/fcimb.2022.932563

7. PLOS authors have the option to publish the peer review history of their article (what does this mean?). If published, this will include your full peer review and any attached files.

Reviewer #1: **Yes: **Ravi Philip Rajkumar

Reviewer #3: No

---

## [Editor Report · Acceptance letter]

28 Mar 2023

PONE-D-22-04448R1 

Demographic, social, and behavioral correlates of SARS-CoV-2 seropositivity in a representative, population-based study of Minnesota residents 

Dear Dr. Beebe:

I'm pleased to inform you that your manuscript has been deemed suitable for publication in PLOS ONE. Congratulations! Your manuscript is now with our production department. 

Kind regards, 

on behalf of

Dr. Siew Ann Cheong 

Academic Editor

PLOS ONE